# A scoping review of "Tang ping" (Lying flat) and mental health status among Chinese youth

**Xinrui Ren** [1]*, **Haslinda Abdullah**[1,2], **Hayrol Azril Mohamed Shaffril**[2], **Haliza Abdul Rahman**[2,3], **Zeinab Zaremohzzabieh**[4]

1 Faculty of Human Ecology, Universiti Putra Malaysia, Serdang, Selangor, Malaysia, 2 Institute for Social Science Studies, Universiti Putra Malaysia, Putra Infoport, Serdang, Selangor, Malaysia, 3 Faculty of Medicine and Health Sciences, Universiti Putra Malaysia, Serdang, Selangor, Malaysia, 4 Women and Family Studies Research Center, University of Religions and Denominations, Qom, Iran

* renxinrui611@gmail.com

## Abstract

"Tang ping" has become a common phenomenon among Chinese youth. The mental health status caused by "Tang ping" has been discussed in many articles, but there has been no scoping review of the research field on "Tang ping" and its mental health status through a complete systematic review procedure. Therefore, this study aims to combine the Preferred Reporting Items for Systematic Reviews and Meta-Analyses Extension for Scoping Reviews (PRISMA-ScR) guidance framework and the Population, Interest, and Context (PICo) framework to conduct a scoping review on "Tang ping" and its mental health status, to help scholars better conduct follow-up research in this field. The review process included following the review protocol, formulating research questions, and a systematic search strategy based on identification, screening, and eligibility. The main databases covered were China National Knowledge Infrastructure (CNKI), Google Scholar, and Scopus. Of 973 articles, 27 articles were selected. The research results indicated that "Tang Ping" is not a single behavioral pattern, but rather a continuum of coping mechanisms for social stress. Mental health outcomes associated with "Tang Ping" include both negative and positive dimensions, with anxiety, depression, and emotional problems being the most common, accompanied by stress relief and emotional regulation. Differences between students and youth workers highlighted the influence of social context. In conclusion, this scoping review demonstrated that "Tang Ping" is closely associated with mental health through accumulated stress, perceived loss of control, and effort-reward imbalance, and lays a structured foundation for future research in this emerging field.

## Introduction

In 2021, "Tang ping" (lying flat) became popular in China. The origin was that a netizen posted an article titled "Tang Ping is Justice" on the Internet to express his lying flat philosophy-life without a stable source of income is also worth promoting

**Data availability statement:** All relevant data are within the paper and its Supporting Information files.

**Funding:** The author(s) received no specific funding for this work.

**Competing interests:** The authors have declared that no competing interests exist.

[1]. With the spread of Internet media and the sharp comments of mainstream media, the concept of "Tang ping" has become controversial. An article published in the mainstream media, Guangming Daily, stated that "Tang ping" is a manifestation of young people unwilling to work hard, choosing to avoid pressure [2]. However, more and more young people want to relax and lie down instead of working hard under the "996" work system, which means employees work 6 days a week, from 9 a.m. to 9 p.m.[3], or in university. They thought "Tang ping" is a low-desire life, a resistance to the unchangeable involution [4], which allowed them to take a breath in the cruel social competition.

"Tang ping" includes characteristics such as resistance, pressure, involution, and low desire. As Ye [5] defined, "Tang ping" is resistance to productivism philosophy, the era of consumerism, capital exploitation, certain unfair social distributions, and the pursuit of subjectivity. In addition, "Tang ping" is indeed a social behaviour reflecting the psychological status of individuals and groups under certain conditions [6]. Furthermore, "Tang ping" is a way of behaviour in which young people refuse to engage in unthinking pseudo-struggles and face life with a low-desire attitude in the face of social pressure and continuous competition under the current background of socio-economic transformation [7].

Although the term "Tang ping" is recent, it can be traced back to its origin. Early, "Sang" is a derogatory term that has the connotation of loss, bad luck, depression, etc.; when "Sang" was introduced into youth culture, it was characterised by self-deprecation, decadence, and a numb way of life [8]. Then "Foxi" (Buddhist) gradually emerged to cope with a high-pressure life. "Tang ping" is a new development of the "Foxi", while the "Foxi" and "Tang ping" are the inheritance of the "Sang" culture to express the resistance to competition [9]. All of these terms present young people's attitudes towards life.

However, there are different opinions about "Tang ping". Some people think that "Tang ping" is harmful to the development of society. The low-desire life advocated by "Tang ping" goes against the traditional Chinese spirit of hard work [9]. This will make young people who should work hard lose their ambition and also suppress their desire to consume. China encourages residents' consumption and establishes and improves a long-term mechanism to expand residents' consumption [10]. In this way, the economy continues to prosper. Additionally, social media plays an important role in shaping young people's behaviours and lifestyles in the digital age [11]. It is not difficult to see this from the spread of the concept of "Tang ping". Shen and Dai [12] worried that students would indulge in a comfortable and relaxed life instead of focusing on their studies and future. Nevertheless, some people think that "Tang ping" can regulate an individual's mental health. According to An et al.[13], some students prefer "Tang ping" to help them release stress and focus on their studies in the future.

From 2021 to 2024, the research on "Tang ping" has continued to increase and has been very popular. By searching for keywords related to "Tang ping", a total of 973 Chinese and English articles were published in the China National Knowledge Infrastructure (CNKI), Google Scholar and Scopus databases between 2021 and 2024.

The research questions of this paper include the following: What are the characteristics of these research articles? What is the methodology they used, and what are the findings? Are there different "Tang ping" types for Chinese youth? Do different "Tang ping" types depend on the Chinese youth's mental health status?

## Methodology

### 2.1 Review protocol

This article followed the Preferred Reporting Items for Systematic Reviews and Meta-Analyses Extension for Scoping Reviews (PRISMA-ScR) guidelines (see S1 File) to review all papers and conduct a scoping review. The PRISMA-ScR, released in 2018, is used for scoping reviews, and its review checklist contains 20 basic reporting items and 2 optional items that need to be included when completing a scoping review [14]. Subsequently, this article formulates research questions based on the Population, Interest, and Context (PICo) framework, which is for qualitative studies [15]. In this study, the population is Chinese youth (P), the phenomenon of interest is "Tang ping" (I), and the context is mental health status (Co). Guided by the PICo framework described above, this scoping review focused on literature examining "Tang ping" among Chinese youth, with mental health as the primary contextual dimension.

According to previous research literature, it was found that researchers' research on "Tang ping" focused on college students' mentality of "Tang ping" and how to deal with college students' "Tang ping" behaviour, providing ideological guidance and proposing strategies for behaviour change. The present review adopted a broader analytical perspective. Specifically, the literature was analysed across several key dimensions, including the general characteristics of included studies, the methodological approaches employed, the conceptualisation and categorisation of "Tang ping", and its reported associations with mental health outcomes. Relevant studies were identified and screened according to pre-defined inclusion and exclusion criteria. Data extraction was conducted in alignment with the analytical framework, and data synthesis was performed using thematic analysis. This methodological approach ensured a systematic and transparent mapping of the existing literature.

### 2.2 Searching studies years

This article searches papers on research on Chinese youth's mental health status of "Tang ping", from 2021 to 2024. In China, "Tang ping" has been very popular since it became a hot word in 2021 [16]. It is very important to study the development status and future trends of this topic, especially its impact on young people.

### 2.3 Systematic searching strategies

**Identification.** Combining the PRISMA-ScR guidance framework and the PICo framework, searched the China National Knowledge Infrastructure (CNKI), Google Scholar, and Scopus databases for all Chinese and international literature related to "Tang Ping" and mental health. CNKI was included primarily to collect empirical research published in Chinese, as the concept of "Tang Ping" is most actively discussed in the Chinese-speaking world. While Google Scholar and Scopus were used to retrieve peer-reviewed international and interdisciplinary research in psychology, social sciences, and related fields.

Consistent with the objectives of a scoping review, this study aimed to map the breadth, key themes, and conceptual development of the existing literature rather than to achieve exhaustive database coverage. The selected databases were therefore considered sufficient to identify major patterns, methodological approaches, and research gaps. In addition, Elicit was used as a supplement to the literature search. The key information for searching the literature is: "Tang ping", "youth", and "mental health". Then, according to the purpose of the scoping review, the search keywords were expanded as much as possible without rigorously assessing the quality of the selected literature. Keywords search is shown in Table 1.

**Table 1. Search strategy in the scoping reviews of the mental health status of "Tang ping" Chinese youth.**

Filter conditions: Published in 2021–2024; Chinese, English

| Step | Database | Query | Search Scope | Result |
|------|----------|-------|--------------|--------|
| #1 | CNKI | "Tang ping" AND (College students OR youth OR young people) AND "Mental health" | Title/Abstract/Keywords | 353 |
| #2 | Scopus | "Tang ping" OR "lying flat" | Title/Abstract/Keywords | 15 |
| #3 | Google Scholar | ("Tang ping" OR "lying flat") AND (College students OR youth OR young people) AND (Mental health OR psychological problem OR psychological well-being OR depression OR anxiety OR burnout OR academic problems OR Internet addiction) | Title/Abstract/Keywords | 598 |
| #4 | Elicit (Additional literature search) | "lying flat" AND "college students" AND "mental health" | | 8 |

**Screening.** With the support of the above research strategy, the inclusion and exclusion criteria were formulated as shown in Table 2. Literature meeting the inclusion criteria includes peer-reviewed articles in both Chinese and English journals. These articles must explicitly explore the concept of "Tang Ping" and examine its empirical relationship with mental health. Studies that do not use "Tang Ping" as a core concept, are unrelated to mental health, lack empirical analysis, or are duplicate publications will be excluded.

**Eligibility.** By reading titles, abstracts, and full texts, 973 documents were screened by following the Preferred Reporting Items for Systematic Reviews and Meta-Analyses Extension for (PRISMA) 2009 Flow Diagram [17]. As shown in Fig 1, to exclude chapters of books, theses, conferences, newspapers, reports, medical, physical education, and other non-social science studies, leave 242 articles. After duplicate articles were eliminated, leaving 149. After screening the titles, 51 articles remain. After screening the abstracts, 46 articles remain. 17 documents were deleted because the content is not focused on "Tang ping" and no data. The full texts of the other two articles could not be downloaded and were deleted after the corresponding authors were emailed and still could not obtain them. The final analysis included 27 articles (see S2 File).

## 2.4 Data extraction and analysis

The data extracted from the above articles included publication year, research region, research design, subjects, and research results (type of "Tang ping", mental health status). To determine the type of "Tang ping", all types in the article were extracted and organised into 4 categories, and all mental health statuses in this article were extracted and organised into 2 categories. In a scoping review, the inclusion of multiple data sources is very important for determining the results of the research topic; the presentation of the included data is also important [18]. To comprehensively consider

**Table 2. Inclusion and exclusion criteria.**

| Feature | Inclusion Criteria | Exclusion Criteria |
|---------|--------------------|--------------------|
| Study Design | Quantitative or qualitative or mixed method papers on "Tang ping" among college student's mental health | Papers that do not focus on college students or "Tang ping" behaviour or mental health status |
| Participants | Chinese youth | Papers that do not include Chinese youth |
| Outcomes | Reports data findings on "Tang ping" | Papers that do not report relevant outcomes |
| Time | From 2021 until 2024 | Outside of 2021–2024 |
| Language | English and Chinese | Papers in other languages |
| Information Sources | CNKI, Google Scholar, Scopus, and Elicit (supplement) | Papers are not in CNKI, Google Scholar, or Scopus |

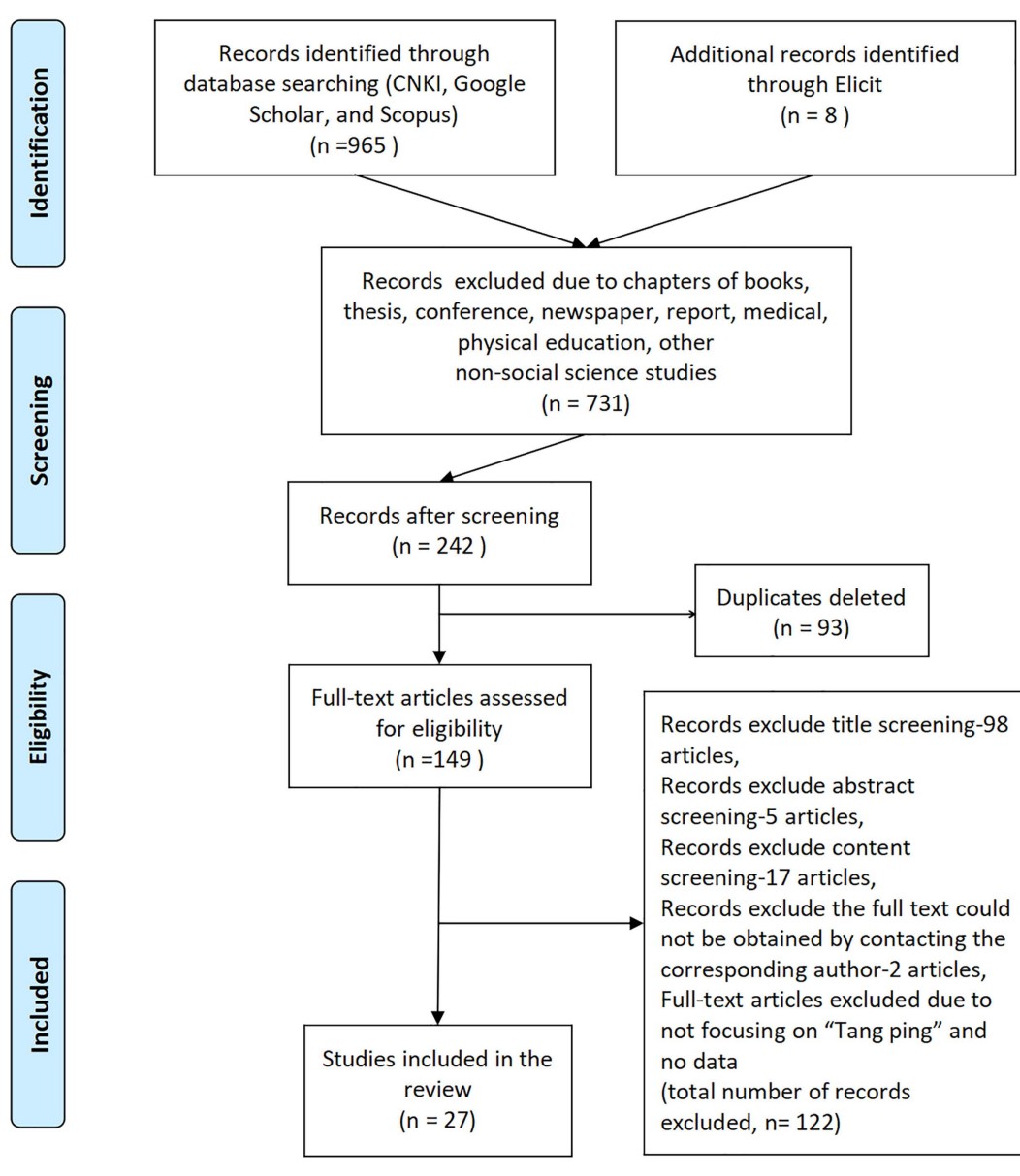

**Fig 1. PRISMA-ScR Flow Diagram of the Search Process.**

diverse research perspectives on "Tang ping" and mental health status, this study did not conduct a quality assessment, consistent with the methodological purpose of scoping reviews.

Based on the collected data, this article was conducted on the publication year, research methods, research subjects, sample region, and the type of "Tang ping" studied, combined with the mental health status of the research subjects.

## 2.5 Ethics statement

This study was approved by the Ethics Committee for Research Involving Human Subjects of Universiti Putra Malaysia (Approval No. JKEUPM-2024–845).

Informed consent was not required, as this study involved a secondary analysis of published literature and did not include direct interaction with human participants.

## Result

### 3.1 Characteristics of studies

Table 3 listed the characteristics of the 27 research articles that were finally included. In terms of research type, quantitative research accounted for 55.6%, qualitative research accounted for 37%, and mixed research accounted for 7.4% (Fig 2). For the publication year, there was no empirical research on "Tang ping" in 2021, while there were 5 articles in 2022 (18.5%), followed by 11 articles in 2023 (40.7%), and 11 articles in 2024 (40.7%). The number of research articles showed an upward trend (Fig 3). In terms of literature language, there were 19 Chinese articles (70.4%) and 8 English articles (29.6%) (Fig 4). Regarding the sample region (Fig 5), 48.3% of articles did not specify which province or city in China

**Table 3. Characteristics of studies included in the review (n = 27).**

| Category | Description | Frequency | Percentage |
|---|---|---|---|
| **Research Type** | Qualitative research | 15 | 55.6 |
| | Quantitative Research | 10 | 37.0 |
| | Mixed Research | 2 | 7.4 |
| **Publication year** | 2022 | 5 | 18.5 |
| | 2023 | 11 | 40.7 |
| | 2024 | 11 | 40.7 |
| **Language** | Chinese | 19 | 70.4 |
| | English | 8 | 29.6 |
| **Sample Region** | Online virtual community | 3 | 10.3 |
| | Not mentioned | 14 | 48.3 |
| | Guangxi | 2 | 6.9 |
| | Beijing | 3 | 10.3 |
| | Zhejiang | 1 | 3.4 |
| | Guangzhou | 1 | 3.4 |
| | Xi'an | 1 | 3.4 |
| | Shandong | 1 | 3.4 |
| | Jiangxi | 2 | 6.9 |
| | Liaoning | 1 | 3.4 |
| **Subjects** | Primary school/ Second high school/ High school students | 2 | 6.5 |
| | College students | 19 | 61.3 |
| | Workers | 4 | 12.9 |
| | Teachers | 2 | 6.5 |
| | Youth | 4 | 12.9 |
| **Type of Tang ping** | Definition | 14 | 51.9 |
| | 2 types | 6 | 22.2 |
| | 3 types | 4 | 14.8 |
| | 4 types | 3 | 11.1 |
| **Mental Health** | Negative Affect | 17 | 77.3 |
| | Positive Affect | 5 | 22.7 |

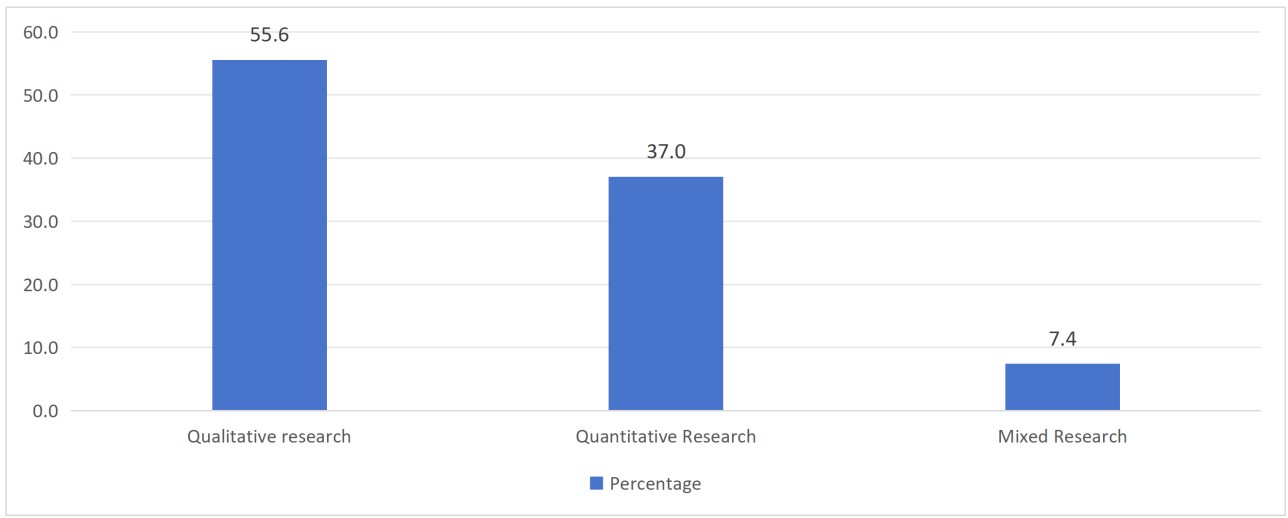

**Fig 2.  The Research Type of Included Articles.**

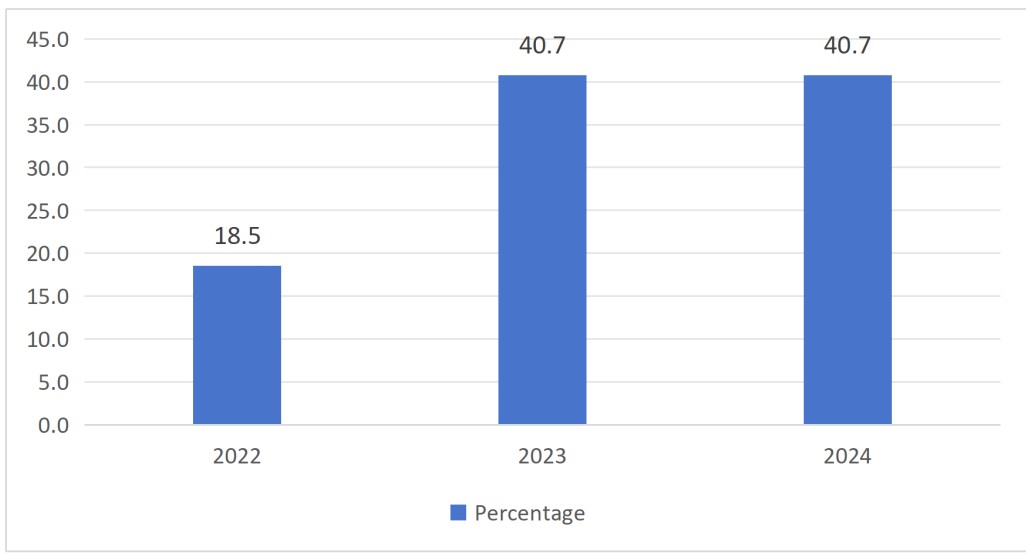

**Fig 3.  The Publication Year of Included Articles.**

the sample came from, followed by the online virtual community (10.3%) and Beijing (10.3%), then followed by Guangxi (6.9%) and Jiangxi (6.9%), and finally Zhejiang, Guangzhou, Xi'an, Shandong, Liaoning, each accounting for 3.4%. For the research subjects (Fig 6), college students accounted for 61.3%, followed by workers (12.9%) and youth (12.9%), and finally primary school or secondary high school or high school students (6.5%) and teachers (6.5%). The research methods will be shown in Table 4. In terms of "Tang ping" types (Table 5), 51.9% of the articles gave a definition, followed by "Tang ping" being divided into 2 categories (22.2%), 3 categories (14.8%), and 4 categories (11.1%). Mental health status related to "Tang ping" can be divided into negative affect (77.3%) and positive affect (22.7%) (Table 6).

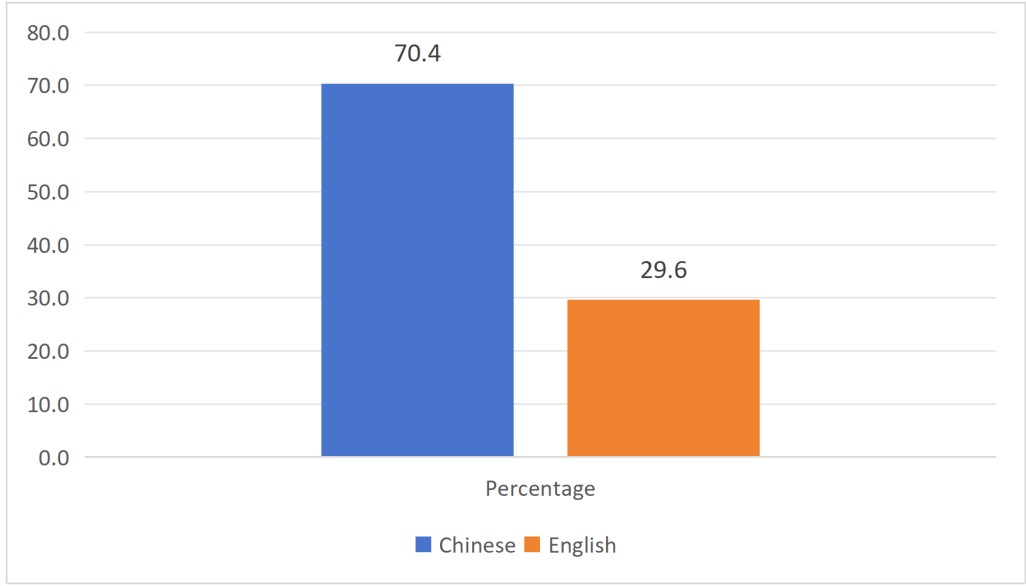

**Fig 4. The Language of Included Articles.**

## 3.2 Categorisation of research method

This study divides the research methods techniques of "Tang ping" into the following three categories: Qualitative method, quantitative method, and mixed method. Qualitative studies primarily relied on interviews, with semi-structured interviews being the most frequently used technique (38.5%), followed by in-depth interviews (30.8%). In addition, 30.8% of qualitative studies reported using interviews without specifying the exact interview format. Quantitative studies predominantly used standardised scales to measure and analyse relevant variables. Among these, the Positive Affect and Negative Affect Scale was the most frequently applied (7.9%), followed by the Patient Health Questionnaire-9, Scenario Questions, Positive Emotions Scale, and Lying-flat Scale (each 5.3%). Mixed-method studies combined qualitative and quantitative techniques, typically incorporating interviews alongside scale-based measures. Table 4 summarises the distributionand frequency of research techniques identified in this review.

## 3.3 Categorisation of "Tang ping" type

By scoping 27 research articles, this article identified different types of "Tang ping" among Chinese youth. The conceptualisation of "Tang Ping" in these studies is mainly reflected in four aspects. Some studies only provide a general definition of "Tang Ping" without further differentiation; others categorise "Tang Ping" into two, three, or four types based on different research purposes or contexts. These different types of "Tang Ping" reflect the operationalisation methods of the concept in different studies. The distribution of these conceptualisations and their corresponding categories is summarised in Table 5.

## 3.4 Categorisation of mental health status

In these articles, the mental health status related to "Tang ping" can be divided into two categories: negative affect and positive affect. Negative affect is divided into 17 categories, including reduced self-worth, anxiety, uncertainty about the future, depression, loneliness, identity crisis among students, emotional problems, boredom and resistance, social withdrawal, lack of initiative, self-blame, learned helplessness, academic burnout, suffering, not working hard, reduced

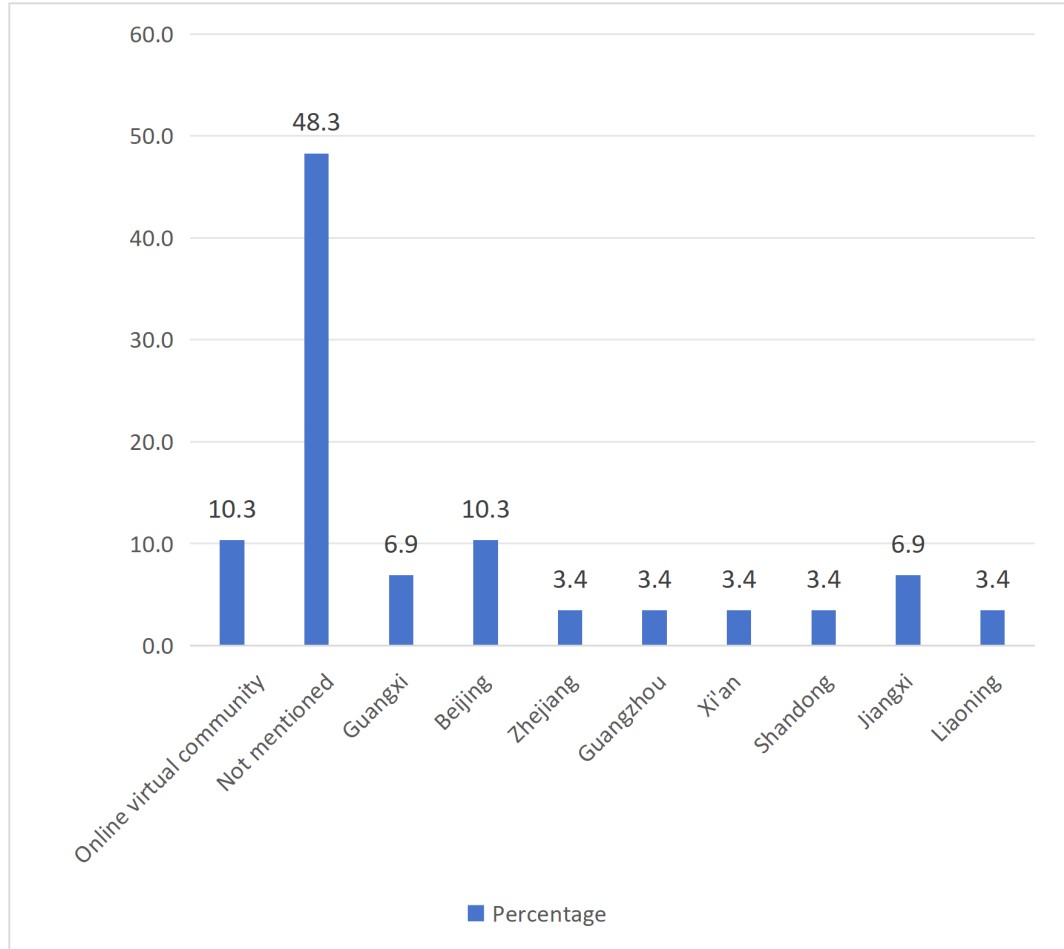

**Fig 5. The Sample Region of Included Articles.**

subjective/overall happiness, and reduced resilience. Among them, anxiety (20.3%) was the most frequently repsorted outcomes, followed by emotional problems (12.5%), and then depression (9.7%). In contrast, the positive effects are divided into five categories: stress reduction, self-adjustment, mental relaxation, anxiety release, and increased psychological well-being. Among them, the most frequent positive mental health outcomes were stress reduction and anxiety release, each accounting for 4.7%. These distributions are summarized in Table 6, indicated that "Tang ping" was associated with both negative and positive mental health outcomes among Chinese youth.

The mental health status of Chinese youth associated with "Tang ping" varies across studies, as illustrated in Figs 7–9. It is worth noting that "Tang ping" young workers lacked self-confidence [19], felt anxiety [20], and had a lower general happiness [7]. Primary school and secondary school, and high school students would experience anxiety and depression [21], accompanied by emotional suppression [22]. Teachers felt confused about their work [23], and burnout led to not working hard [24]. However, "Tang ping" is also associated with positive mental health. For some teachers, young workers and college students, "Tang ping" is a way to relax from stress (13,24,25), avoid burnout [24], and a way to self-regulate to better cope with the next study and work [13,23]. "Tang ping" college students have almost all negative and positive mental health statuses.

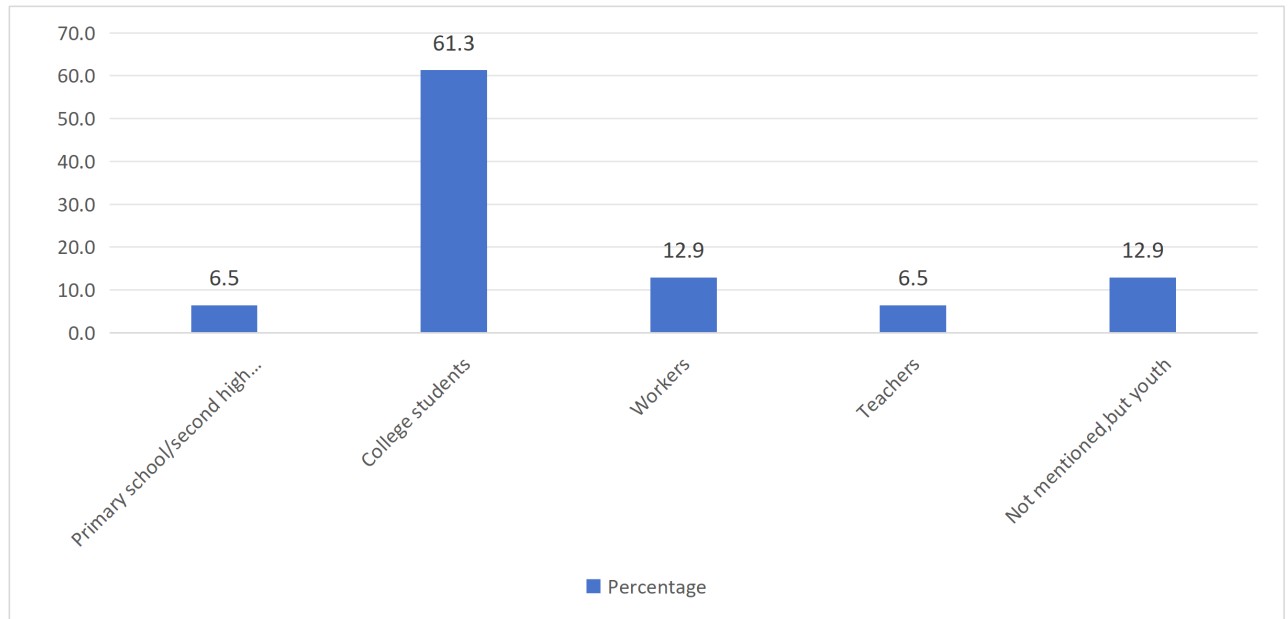

**Fig 6. The Subjects of Included Articles.**

## Discussion

### 4.1 Factors affecting "Tang ping"

After COVID-19, the hot word "Tang ping" has expanded and exploded rapidly [25]. It can be said that 2021 is the first year of "Tang ping" in China. The emergence of "Tang ping" is influenced by factors such as society, individuals, family, and social media.

First of all, the society is seriously "involution". The fierce social competition and huge social pressure make young people feel that it is difficult to get corresponding rewards even if they work hard, which leads to a sense of helplessness and fatigue [7,13,20,21,23,24,26–36]. They want to escape the pressure and choose to "Tang ping".

Secondly, personal psychological factors. Some young people may have mental problems such as learned helplessness, inferiority, and low self-efficacy, which lead to their loss of value and lack of motivation and confidence to struggle [21,27,30,33–35,37–43].

Next, a lack of social support causes young people to "Tang ping". This is mainly manifested in the fact that young people lack sufficient understanding and support from their families, friends, and schoolmates, which makes them feel isolated and helpless when facing pressure [21,29,34,35].

Finally, the influence of social media causes young people to "Tang ping". Some young people may be influenced by the Internet and indulge in the virtual world [39], following the trend of "Tang ping" [19–21,24,29,37]. Dai et al. [21]believed that young people with high levels of anxiety and depression will also have high levels of Internet addiction and "Tang ping" behaviour.

### 4.2 Definition of "Tang ping"

Literature reviews indicated that there was currently no unified definition for "Tang ping", but rather a variety of interpretations, some even overlapping. "Tang ping" originated online and exhibited rich connotations and extensions. It evolved from a colloquial expression in online youth culture to being incorporated into academic discourse [19,32,40]. Most studies

**Table 4. Categorisation of research method.**

| Category | Subcategory (Research Technique) | Frequency | Percentage | No. of Articles |
|---|---|---|---|---|
| Qualitative Research Method | Semi-structure interview | 5 | 38.5 | [13,19,30,32,34] |
| | In-depth interview | 4 | 30.8 | [27,28,37,59] |
| | Not mentioned | 4 | 30.8 | [23,29,31,33] |
| Quantitative Research Method | Positive Affect and Negative Affect Scale (PANAS) | 3 | 7.9 | [36,40,45] |
| | Patient Health Questionnaire-9 (PHQ-9) | 2 | 5.3 | [21,43] |
| | Scenario Questions (Self-constructed) | 2 | 5.3 | [35,45] |
| | Positive Emotions Scale (PES) | 2 | 5.3 | [36,40] |
| | Lying-flat Scale (LfS-12) | 2 | 5.3 | [7,41] |
| | Adolescent Involution and Lying Flat Scale (AILFS-20)(Self-constructed Questionnaire) | 1 | 2.6 | [21] |
| | Investigation of "Lying flat" Youth Phenomenon (Self-constructed Questionnaire) | 1 | 2.6 | [39] |
| | General Anxiety Disorder-7 (GAD-7) | 1 | 2.6 | [21] |
| | Difficulties in Emotion Regulation Scale(DERS-16) | 1 | 2.6 | |
| | Negative Cognitive Processing Bias Questionnaire (NCPBQ-15) | 1 | 2.6 | |
| | Internet Addiction Test (IAT-20) | 1 | 2.6 | |
| | Digital media ascending social comparison scale | 1 | 2.6 | [19] |
| | Digital media use dependence scale | 1 | 2.6 | |
| | Lying flat scale (Self-constructed Questionnaire) | 1 | 2.6 | |
| | Beliefs of Effort Scale (BES-8) | 1 | 2.6 | [35] |
| | Lying flat Tendency Scale (LFTS-6) | 1 | 2.6 | [45] |
| | Marlowe Crowne Social Desirability Scale (MCSD) | 1 | 2.6 | |
| | Happiness Index Scale (HIS-9) | 1 | 2.6 | |
| | Basic Psychological Need Satisfaction Scale (BPNS-21) | 1 | 2.6 | |
| | Satisfaction with Life Scale (SWLS) | 1 | 2.6 | [40] |
| | Cognitive Biases Scale (CBS) | 1 | 2.6 | |
| | College Students' Attitude Towards Layflat Psychology Scale (CSATLPS-15)(Self-constructed Questionnaire) | 1 | 2.6 | |
| | Mental Toughness Scale (MTS) | 1 | 2.6 | [36] |
| | Self-efficacy Scale(SeS) | 1 | 2.6 | |
| | Comprehensive Happiness Questionnaire (CHQ) | 1 | 2.6 | [41] |
| | Positive Psychological Capital Questionnaire (PPQ-26) | 1 | 2.6 | [42] |
| | Shandong Province Youth Health Subject Database | 1 | 2.6 | [22] |
| | Lying Flat Behaviors Questionnaire (LFBQ-9)(Self-constructed Questionnaire) | 1 | 2.6 | [43] |
| | General Well-Being Scale (GWB-18) | 1 | 2.6 | [7] |
| | Self-esteem Scale (SeS-10) | 1 | 2.6 | |
| | Achievement Motivation Scale (AMs-30) | 1 | 33.3 | |
| | Questionnaire on the Phenomenon of "Lying flat" among Young People | 1 | 2.6 | [20] |

viewed "Tang ping" as a phenomenon or behaviour that deviated from mainstream culture and confronted the current social structure. Just as Su [44] mentioned, "Tang ping" is a spontaneous resistance to social inequality and a collective and desperate yearning for social change. Meanwhile, young people's tendency to "Tang ping" may be accompanied by feelings of helplessness and low-desire [26,30]. They may also relieve psychological stress through short-term psychological adjustment [3,8,12–14,21]. It may also become a coping strategy to avoid effort and resist stress [20,22,35,36,39,42,45,46].

**Table 5. Categorisation of "Tang ping" type.**

| Category | Subcategory | No. of Articles |
|---|---|---|
| **Definition** | Low-desire Tang ping type, characterized by withdrawal, low ambition, self-isolation, and emotional numbness. | [29] |
| | "Tang ping" is just a self-joking way for young people to release their emotions, and it is a self-protection and defence mechanism. | [32] |
| | "Learning Tang Ping" includes four patterns:1. Learned helplessness, 2. Low self-efficacy, 3. Escapist-compensatory behavior, 4. Self-mockery and ambivalence. | [38] |
| | "Tang ping" means giving up the pursuit of a high salary and high position and choosing a low-consumption, low-pressure, and low-demand lifestyle. | [59] |
| | Academic "Tang ping" | [21] |
| | "Tang ping" is described as a state of lacking willpower and escapism. | [39] |
| | "Tang ping" expresses a new attitude towards life in most contexts. | [19] |
| | "Tang ping" is negatively resistant behaviour toward social competition; a simple lifestyle without effort making. | [35] |
| | "Tang ping" refers to the state of some youths who choose to give up their efforts and passively escape when the pressure they are under breaks through the individual's psychological threshold. | [45] |
| | "Tang ping" refers to the behavioral attitudes of university students that show negative idleness and resistance to mainstream values. | [40] |
| | The psychological phenomenon of "Tang ping" as a kind of avoidance coping strategy when individuals face life pressure. | [36] |
| | "Tang ping" is a mental or behavioral state in which young people choose to give up efforts and passively escape when the pressure they are under exceeds their psychological critical value. | [42] |
| | "Tang ping" is a conservative emotion regulation strategy adopted by young people to suppress their emotions. | [22] |
| | The "Tang ping" phenomenon is not the mainstream mentality of young people, but a short-term strategy for young people to cope with pressure. | [20] |
| **2 Types** | Complete "Tang ping" (passive and negative), Intermittent "Tang ping" (active and positive). | [13] |
| | Passive "Tang ping" (withdrawal, demotivation), Rational "Tang ping" (self-protection, reflection) | [30] |
| | Passive withdrawal, Selective effort | [31] |
| | 1. Negative life events (illness, unemployment, etc.) force people to "Tang ping". 2. Modern social pressure causes people to "Tang ping": Withdrawing "Tang ping", Ritualistic "Tang ping", Temporary "Tang ping". | [34] |
| | "Escape" Tang ping, "Awakening" Tang ping | [41] |
| | Awakening "Tang ping", Avoiding "Tang ping" | [7] |
| **3 Types** | Passive resignation "Tang ping", Strategic withdrawal "Tang ping", Mental disengagement "Tang ping" | [27] |
| | Passive "Tang ping" (withdrawal), Active "Tang ping" (reflective, adaptive retreat), Strategic compromise | [23] |
| | Evasive behaviours, Emotional compensation, Swan strategy | [24] |
| | Academic "Tang ping", Life "Tang ping", Social "Tang ping" | [43] |
| **4 Types** | 1. "45 degrees" active balance tendency 2. "45 degrees" passive response 3. "90 degrees" sprint and "45 degrees" rest 4. "90 degrees" and "0 degrees" dilemma | [37] |
| | Negative withdrawal from work, Emotional detachment, Lack of motivation to learn, and A life attitude without desires or expectations | [28] |
| | Reclusive "Tang ping", Falling "Tang ping", Adjusting "Tang ping", and Semi-formed "Tang ping" | [33] |

Some literature further subdivided "Tang ping" into different types to facilitate measurement, intervention, or policy discussion. It is usually regarded as a behavioural tendency that deviated from the mainstream culture and opposes struggle and competition [28,37,43,47], to capture the coping methods of Chinese youth when facing different psychological states while "Tang ping". For example, the proactive, rational, selective, or intermittent types of "Tang ping" indicated that it may be a temporary and adaptive psychological adjustment that enabled individuals to regulate stress and restore emotional

**Table 6. Categorisation of mental health status.**

| Category | Subcategory | Frequency | Percentage | No. of Articles |
|---|---|---|---|---|
| **Negative Affect** | Self-doubt/low self-worth/Confusion/Lack of self-confidence | 5 | 7.8 | [19,24,27,29,38] |
| | Anxiety | 13 | 20.3 | [13,20,21,23,27–29,31–34,37,39] |
| | Uncertainty about the future | 2 | 3.1 | [27,59] |
| | Depression | 6 | 9.4 | [21,23,28,29,37,43] |
| | Loneliness | 1 | 1.6 | [37] |
| | Identity crisis among students | 1 | 1.6 | |
| | Emotional problem: Emotional exhaustion/Apathy/Emotional numbness/Emotional contradiction/Negative emotion/Emotional suppression | 8 | 12.5 | [22,28,29,31,36,38,40,45] |
| | Boredom/Annoyance/Disgust/Disappointment/Resistance | 1 | 1.6 | [29] |
| | Social withdrawal | 2 | 3.1 | [29,30] |
| | Lack of initiative | 1 | 1.6 | [30] |
| | Shame/Self-blame | 2 | 3.1 | [34,39] |
| | Learned helplessness | 1 | 1.6 | [38] |
| | Academic burnout | 3 | 4.7 | [38,43,59] |
| | Suffering | 1 | 1.6 | [39] |
| | Not working hard | 3 | 4.7 | [24,42,45] |
| | Reduce subjective well-being/General happiness | 2 | 3.1 | [7,41] |
| | Reduce resilience | 1 | 1.6 | [42] |
| **Positive Affect** | Stress reduction | 3 | 4.7 | [13,24,59] |
| | Self-adjustment | 2 | 3.1 | [13,23] |
| | Mental relaxation | 2 | 3.1 | [13,59] |
| | Relieve anxiety/Relief | 3 | 4.7 | [30,38,59] |
| | Psychological well-being | 1 | 1.6 | [41] |

balance [7,13,29,31,38,48]. In addition, "Tang ping" may reflected feelings of detachment, helplessness or withdrawal, which are often caused by repeated setbacks, limited psychological resources or structural disadvantages [30,34,48–50]. Between these two extreme situations, some studies describe strategic or selective forms of "Tang ping" in order to seek a middle ground and achieve reconciliation with the self [23,24,27]. Therefore, current researches cannot simply classified "Tang ping" into a specific type.

However, rather than saying that there is a conceptual inconsistency in "Tang ping", it is more accurate to say that the diversity of definitions reflects the complexity of "Tang ping" as a multifaceted phenomenon, which is rooted in a specific Chinese socio-cultural context. Scholars have conducted extensive research on this phenomenon from the fields of sociology, psychology, education and adolescent studies [20,21,28,43]. This conceptual diversity and complexity highlighted the necessity of constructing an integrated framework for the concept of "Tang ping". It also emphasised that "Tang ping" is not a single, unified phenomenon, but rather a complex structure constrained and influenced by individual agency, mental state, context, social expectations, and social pressures.

Importantly, the variation in the number of proposed "Tang ping" types did not necessarily indicate theoretical or research methods inconsistency, but rather reflected the exploratory nature of this emerging phenomenon. As research on "Tang ping" continues to develop, future studies exploring the relationship between different types of "Tang ping" and the mental health status among Chinese youth should focus on integrating more unified operational concepts.

**Fig 7.  The Mental Health Status of "Tang ping" Young Workers.**

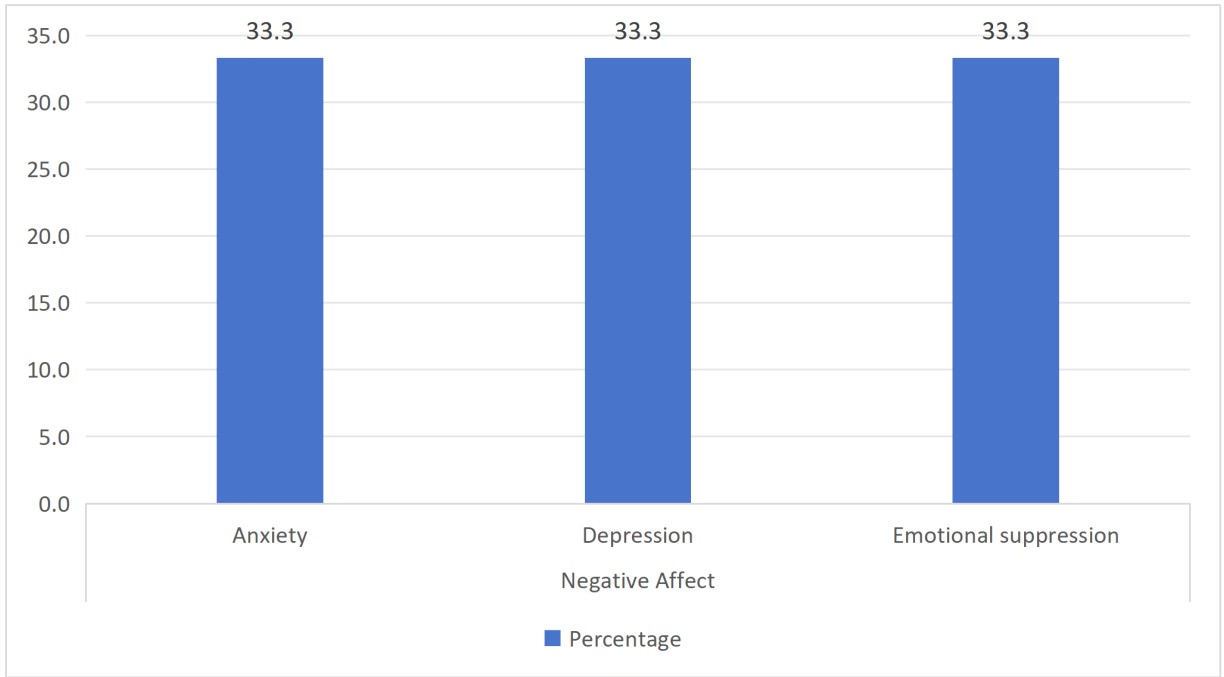

**Fig 8.  The Mental Health Status of "Tang ping" Primary School/Second High School/High School Students.**

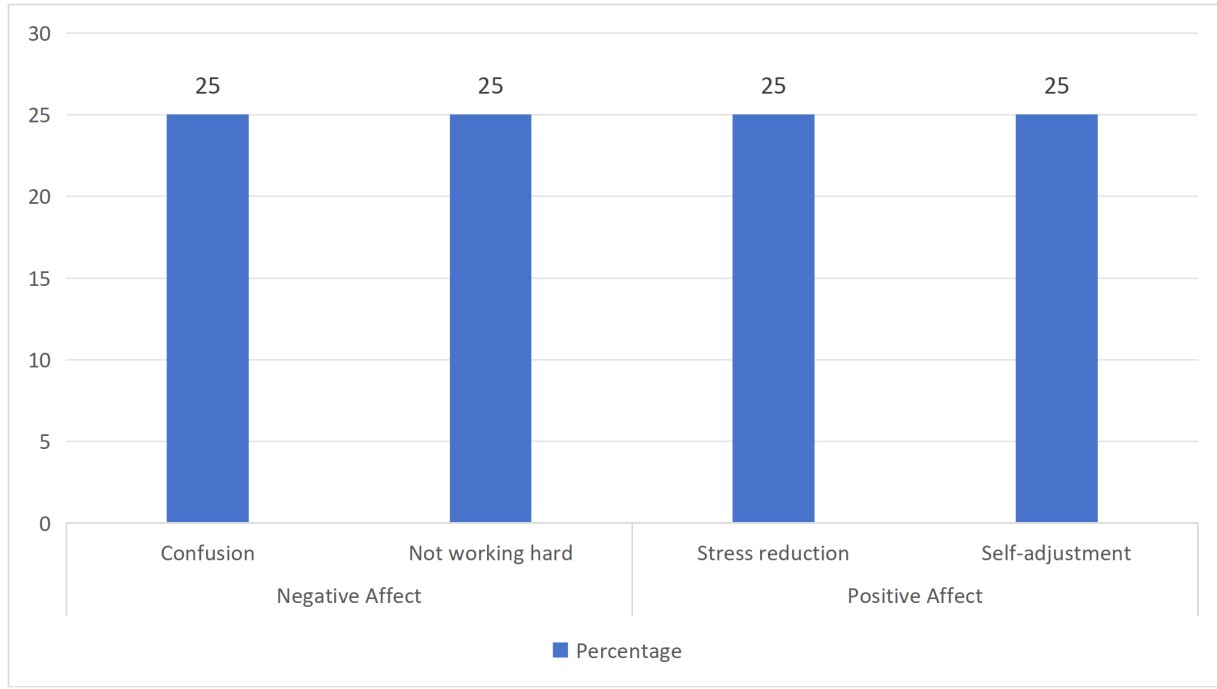

**Fig 9. The Mental Health Status of "Tang ping" Teachers.**

## 4.3 Research methods and future trends of "Tang ping"

The number of research articles on "Tang ping" has increased over the past four years. It is worth noting that the number of articles on empirical research on "Tang ping" has also increased, reflecting the growing interest in "Tang ping". However, 85% (23/27*100%) of the articles analysed in this study were cross-sectional studies, using a variety of psychological scales to measure "Tang ping" and other variables. Two articles conducted intervention studies by using AI algorithms and psychological counselling services [36,40]. he other two papers were about the development of the "Tang ping" Scale [7,45]. These studies fully demonstrated the depth of scholars' and society's attention to "Tang ping". In the future, more intervention and longitudinal studies should be carried out to more comprehensively measure the impact of "Tang ping".

## 4.4 Mental health status of "Tang ping" Chinese youth

The United Nations has proposed 17 goals for the 2030 Sustainable Development Goals, the third of which is to "ensure healthy lives and promote well-being for all at all ages" [51]. The generation of "Tang ping" affects the realisation of this goal. According to the research in this article, young people will have various mental health conditions when they "Tang ping". Negative mental health conditions are mainly concentrated in anxiety, depression, and emotional problems, and the age range is from 7 to 36 years old. Moreover, UNICEF China reported that almost 25% of adolescents felt mild or severe depression and estimated that at least 30 million children and adolescents under 17 years of age in China struggle with emotional or behavioural problems [52]. It claimed that psychological problems are becoming younger age.

**4.4.1 Negative mental health status.** Among young people of different ages and social roles in China, the anxiety, depression, and emotional problems caused by "Tang Ping" are the most frequently mentioned mental health issues in the literature [21–24,32,34,45]. In adolescents, academic pressure and emotional repression are often considered to be the

triggers for anxiety and depression [21,22,43,53]. This explanation is consistent with evidence that mental health problems among Chinese adolescents are occurring at increasingly younger ages [52].

For college students, "Tang ping" behaviour is usually associated with cumulative pressures such as academic stress, job uncertainty, family expectations, and peer comparisons [20,23,28,34]. When multiple sources of stress converge, and the upward flow path is unclear, detachment from reality may become a psychological protection strategy [20,34]. Therefore, college students' "Tang ping" may represent a temporary retreat aimed at conserving emotional resources, rather than a permanent abandonment of effort.

Studies of teachers and youth workers further emphasised the role of effort-reward imbalance in shaping "Tang ping" behaviour [23,24,35,54]. When sustained effort failed to bring expected rewards such as income, promotion, or social recognition, individuals may feel helpless, and their subjective well-being may decrease [23,35,54]. Furthermore, not only does "Tang ping" bring with it psychological pressures such as self-blame and guilt [34], but young people also face tremendous life pressure. In Zhou's study [55], the housing price in China's first-tier cities is 20 times the national average wage, and young people become "Tang ping" in despair after recognising the reality. In this context, "Tang Ping" became a form of passive resistance or self-protection, used to combat perceived structural injustice [33,55,56].

This pattern indicates that "Tang ping" is not an age-specific phenomenon, but a cross-contextual psychological response to shared social pressures [32,45]. From a psychological perspective, continuous exposure to uncontrollable stressors can weaken self-efficacy and exacerbate emotional exhaustion, which may explain the convergence of anxiety and depressive symptoms among different "Tang ping" groups [34]. In this sense, "Tang ping" can be further interpreted as a stress regulation response, rather than just a manifestation of individual vulnerability.

In conclusion, these findings suggested that the negative mental health consequences associated with "Tang ping" should not be understood merely as individual psychological disorders. They represented a collective stress response to structural constraints within a broader socioeconomic and institutional context. This response is driven by the chronic stress, limited opportunities, and declining expectations of social mobility experienced by young people in China [34–36,45,56].

In addressing the negative psychological problems caused by "Tang ping", parents and teachers need to pay attention to students' behaviour and provide stable social support to help students develop mental health [57]. Secondly, schools need to open up promotion standards for teachers with different teaching experiences. Forson et al. [58] believed that salary, work environment, and performance management system can all motivate teachers. Zhou and Wu [33] also suggest providing a "struggle-motivated" work environment for young workers.

**4.4.2 Positive mental health status.** On the contrary, the positive mental health status produced by "Tang ping" can help young workers to self-regulate, reduce stress, and face work and the future with a more positive mental attitude [24,59]. For all college students, in addition to these two points, "Tang ping" can also help them with mental relaxation [38,59], relief of anxiety [30,38,59], and psychological well-being [41]. Moreover, "Tang ping" can allow students to have a double baptism of mental and physical after a period of "involution", to prepare for the next hard study [25].

In this study, the negative mental health status of "Tang ping" among young people is more categories and has a greater impact than the positive effects. Moreover, the mental health status of different groups of "Tang ping" young people is also different. Therefore, in the future, it should pay attention to the negative mental health effects of "Tang ping" on adolescents. In particular, greater emphasis should be placed on how students should promote their mental health development, and how to deal with anxiety, depression and emotional problems that accompany "Tang ping". For young workers, special attention needs to be paid to how to help them regain their confidence, reduce stress, and then work hard to avoid "Tang ping". Only through such a sustainable development concept, continuous attention, and helping young people deal with the various negative mental health effects that "Tang ping" may bring, can we make progress in achieving human mental health at all ages.

Although this study reviewed "Tang ping" and mental health status among Chinese youth, it still has the following limitations. First, the search scope of this study did not cover unpublished research articles, and not all databases are included in the search results. Second, since only articles written in Chinese and English were included, this limits the possibility of generalising the findings to non-Chinese and non-English speaking countries. Third, in terms of quality assessment, to fully consider different opinions, it did not conduct a quality assessment of the papers included in this study. Future studies that can address these limitations should be conducted.

## 5. Conclusion

This study used a scoping approach to review research papers that explored "Tang ping" and mental health status among Chinese youth. From 2021 to 2024, 27 papers were selected from 973 articles published on this research topic based on the screening criteria of this study and included and reviewed. The results showed that people's interest in "Tang ping" and mental health status is growing, and there are differences in "Tang ping" and mental health status among different types of subjects.

This scoping review revealed the significant diversity in the concepts, methodologies, and empirical evidence of the emerging field of "Tang ping". A key contribution of this review lies in demonstrating that "Tang ping" is not a singulator fixed construct, but a multidimensional and context-dependent phenomenon that has been variously interpreted as disengagement, coping, or resistance in response to social and psychological pressures. Meanwhile, "Tang ping" also has certain positive psychological effects. By mapping these heterogeneous conceptualisations alongside reported mental health outcomes this review provides a clearer framework for understanding the complexity of "Tang ping" and highlights the need for clearer conceptual integration in future research. Another important finding of this study is that the negative impact of "Tang ping" is gradually getting younger. Adolescents are more likely to have mental problems than before. Compared with the survey in 2008, the depression problem of young people in 2019 increased by 5.3%, and the main sources of stress for young people are economic pressure, career confusion, and academic pressure [60]. Looking back at the papers in this study, to fight and relieve these pressures, adolescents choose to "Tang ping", and the negative mental health status that accompanies "Tang ping" made students less willing to go to school, have worse academic resilience, and have a weaker sense of learning efficacy [61]. From mental problems to "Tang ping" and then to mental problems, a vicious cycle has been formed.

Therefore, it is necessary to strengthen mental health education in schools and provide psychological counselling services so that adolescents can get effective help when they have mental health problems. Moreover, it is necessary to improve students' psychological capital, including self-efficacy, resilience, hope and optimism [42], so that they are more confident and capable of dealing with setbacks and challenges. For young workers, the improvement of psychological capital may reduce occupational stress and burnout levels [62]. A study of 316 English teachers in China showed that the improvement of psychological capital helps to improve their work commitment and thus achieve results in work [63].

Additionally, this study found that young workers generally "Tang ping" because their efforts are not proportional to their rewards, and they are seriously involved [24,33,35,45]. In China, under the "996" work system in recent years, or even the "007" work system (working from 0:00–0:00, 7 days a week), or "715" work systems (starts at 9:30 a.m. and ends at 12:30 p.m., with a total of 15 working hours and 7 days a week), the work pressure of young people is increasing day by day, and their physical and mental health is difficult to guarantee [3,64]. Given this, work units need to establish a fair and reasonable promotion mechanism, provide competitive salaries and benefits, and provide training and development opportunities to enhance the initiative of young workers. Young people should improve their cognition and learn to regulate emotions for coping with stress [28,40,65]. The state and society should strengthen the protection of the legitimate rights and interests of young workers, create more employment opportunities and development space [20,59], provide humanistic care [24], promote the spirit of struggle, and avoid "Tang ping".

Furthermore, it found that young people are exposed to and influenced by online media. The frequent use of online media makes all information easily exposed to everyone, and people are very likely to receive misinformation related to health [66]. In addition, people with lower education levels are more likely to make social comparisons with digital media, become dependent on digital media, and are more likely to be persuaded or influenced by the groups they interact with daily [19], and have value identification [23], thus developing a tendency to "Tang ping". The Blue Book on Mental Health, released by China shown that the average daily short video usage time for adolescents exceeds 90 minutes, that for college students is nearly 180 minutes, and that for working adults is nearly 140 minutes; high-intensity use is significantly associated with the risk of depression and anxiety [67]. This shows that excessive addiction to online media among young people will increase the risk of psychological problems. It is necessary to carry out online safety education among young people, carefully identify the authenticity of online information, and strengthen the real social support system.

## Supporting information

**S1 File. PRISMA-ScR checklist.**
(PDF)

**S2 File. List of included studies and screening data.**
(XLSX)

## Author contributions

**Conceptualization:** Xinrui Ren, Haslinda Abdullah, Hayrol Azril Mohamed Shaffril.

**Data curation:** Xinrui Ren.

**Formal analysis:** Xinrui Ren.

**Investigation:** Xinrui Ren.

**Methodology:** Xinrui Ren, Hayrol Azril Mohamed Shaffril.

**Project administration:** Xinrui Ren.

**Supervision:** Haslinda Abdullah, Haliza Abdul Rahman, Zeinab Zaremohzzabieh.

**Visualization:** Xinrui Ren.

**Writing – original draft:** Xinrui Ren.

**Writing – review & editing:** Xinrui Ren, Haslinda Abdullah, Hayrol Azril Mohamed Shaffril, Haliza Abdul Rahman, Zeinab Zaremohzzabieh.

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
