## [Decision Letter · Decision Letter 0]

4 Dec 2025

PONE-D-25-38209A Scoping Review of “Tang Ping” (Lying flat) and Mental Health Status on Chinese YouthPLOS ONE

Dear Dr. Ren,

Thank you for submitting your manuscript to PLOS ONE. After careful consideration, we feel that it has merit but does not fully meet PLOS ONE’s publication criteria as it currently stands. Therefore, we invite you to submit a revised version of the manuscript that addresses the points raised during the review process.

We look forward to receiving your revised manuscript.

Kind regards,

Jisheng Liu, Ph.D.

Academic Editor

PLOS ONE

Comment from Journal Office: The comments provided by Reviewer 3 are not relevant to this submission and should be disregarded.

Journal Requirements:

3. Thank you for providing your underlying data as Supporting Information.

We note that the data set contains text or data that is not in English. Please note that PLOS is an English-language publisher, so we require data sets to be provided in English as well. Please upload an English-language version of your data set.

This will also allow us to determine if your data follows PLOS standards per our Data Availability policy here: https://journals.plos.org/plosone/s/data-availability

Reviewers' comments:

Reviewer's Responses to Questions

**Comments to the Author**

1. Is the manuscript technically sound, and do the data support the conclusions?

Reviewer #1: Partly

Reviewer #2: Yes

Reviewer #3: Yes

Reviewer #4: Yes

2. Has the statistical analysis been performed appropriately and rigorously? 

Reviewer #1: N/A

Reviewer #2: I Don't Know

Reviewer #3: Yes

Reviewer #4: Yes

3. Have the authors made all data underlying the findings in their manuscript fully available?

Reviewer #1: No

Reviewer #2: Yes

Reviewer #3: Yes

Reviewer #4: Yes

4. Is the manuscript presented in an intelligible fashion and written in standard English?

Reviewer #1: Yes

Reviewer #2: Yes

Reviewer #3: Yes

Reviewer #4: No

5. Review Comments to the Author

Reviewer #1: PLOS ONE Reviewer Comments

1. Validity of the Study

The manuscript follows a scoping review methodology and refers to PRISMA-ScR, which is appropriate for the topic. The inclusion of CNKI ensures coverage of Chinese-language studies, which is important. However, the search was limited to three main databases, which may restrict its comprehensiveness. The absence of a quality assessment is acknowledged, but a clearer justification strengthens validity. Overall, the study is methodologically sound, although transparency in reporting could be improved in the Methods and Results sections.

2. Originality and Contribution to the Field

The topic of “Tang Tang is highly original and culturally significant. To my knowledge, this is the first scoping review to synthesize evidence on “Tang ping” and mental health status in Chinese youth. This paper brings together different conceptualizations of “Tang ping” and links them to both negative and positive mental health outcomes. This study makes a meaningful contribution to youth studies, psychology, and cross-cultural social sciences.

3. Quality of Presentation

The manuscript is generally well structured but can benefit from language editing to improve clarity and readability. The results section is somewhat lengthy and repetitive; summarizing findings in tables or diagrams (e.g., categorization of “Tang ping” types, positive/negative mental health outcomes) would enhance accessibility. The figures and tables in the current draft appear incomplete or not fully formatted and should be revised before publication.

4. Importance of Findings to the Community

The findings are relevant not only for Chinese social and mental health research but also for understanding broader global youth responses to social pressure and overwork. This study has implications for policymakers, educators, and health professionals concerned with youth well-being. It also highlights the need for more longitudinal and interventional research, which is an important message for the scholarly community.

5. Specific Comments for Improvement

• Clarify the primary research question(s) in the Introduction to ensure consistency.

• We expanded slightly on why databases beyond CNKI, Scopus, and Google Scholar were not included.

• Provide justification for the decision to not conduct a formal quality assessment of the included studies.

• Consider reorganizing the results to avoid repetition, possibly using summary tables or schematic diagrams.

• The abstract should be shortened to emphasize key findings and implications, rather than methodological details.

• Revise figures and tables to ensure clarity, captions, and correct formatting.

• A light copyedit to smooth grammar and simplify long sentences would improve the readability.

Overall Recommendation:

This valuable and timely study. Revisions focused on presentation and clarity could make a strong contribution to PLOS ONE. I recommend minor to moderate revision.

Reviewer #2: I think the book examines in detail the Chinese phenomenon of Tang Ping, which is similar but not identical to Japan's hikikomori.

Personally, I was also intrigued by the proportion of cases where the underlying condition is a developmental disorder and the negative state develops secondary to it.

Reviewer #3: The topic is important and the authors present useful findings. However, I recommend minor revisions to strengthen clarity and transparency.

First, the Ethics Statement should explicitly include the institutional review board approval number and full committee name. Although ethical approval is mentioned, the approval number is missing from the manuscript text, which is required for compliance with PLOS ONE guidelines. Please insert the specific approval identifier rather than referencing it only in supplementary correspondence.

Second, the description of informed consent needs clearer detail. The manuscript states that consent was obtained, but it does not clarify whether written, verbal, or digital consent was used or how confidentiality was maintained. A brief explanation of the consent documentation process and protection of participant identity will improve transparency.

Third, the Methods section would benefit from clarifying the sampling frame and eligibility criteria. The study population is described, but the inclusion/exclusion process is not fully documented. Adding exact criteria and a brief statement on data completeness will enhance reproducibility.

Overall, these adjustments are minor but essential for full compliance with journal standards.

Reviewer #4: The abstract is well-written and effectively summarizes the study. However, it requires revision, particularly in the findings section, where excessive detail should be condensed to improve clarity and conciseness.

Additionally, the introduction needs to be restructured to enhance logical coherence, as the flow between sentences is currently unclear in several parts.

6. PLOS authors have the option to publish the peer review history of their article (what does this mean? ). If published, this will include your full peer review and any attached files.

**Do you want your identity to be public for this peer review?** For information about this choice, including consent withdrawal, please see our Privacy Policy .

Reviewer #1: No

Reviewer #2: No

Reviewer #3: No

Reviewer #4: No

---

## [Author Response · Author response to Decision Letter 1]

12 Jan 2026

Jisheng Liu, Ph.D.

Academic Editor

PLOS ONE

---

## [Editor Report · Decision Letter 1]

18 Jan 2026

PONE-D-25-38209R1A Scoping Review of “Tang Ping” (Lying flat) and Mental Health Status on Chinese YouthPLOS One

Dear Dr. Ren,

Thank you for submitting your manuscript to PLOS ONE. After careful consideration, we feel that it has merit but does not fully meet PLOS ONE’s publication criteria as it currently stands. Therefore, we invite you to submit a revised version of the manuscript that addresses the points raised during the review process.

We look forward to receiving your revised manuscript.

Kind regards,

Jisheng Liu, Ph.D.

Academic Editor

PLOS One

Additional Editor Comments:

The manuscript has been revised in response to the reviewers’ comments, and responses have been provided accordingly. However, several minor details still require further refinement:

1) Abbreviations: All abbreviations must be spelled out in full at their first occurrence, followed by the abbreviated form in parentheses, such as CNKI.

2 )Reference Formatting: The reference list requires consistent formatting. For instance, some entries use authors’ full names, while others use a mix of full surnames and abbreviated given names. Please ensure uniformity according to PLoS One’s style guidelines.

---

## [Author Response · Author response to Decision Letter 2]

25 Jan 2026

Jisheng Liu, Ph.D.

Academic Editor

PLOS One

---

## [Editor Report · Decision Letter 2]

26 Jan 2026

A Scoping Review of  "Tang Ping" (Lying flat) and Mental Health Status on Chinese Youth

PONE-D-25-38209R2

Dear Ren,

We’re pleased to inform you that your manuscript has been judged scientifically suitable for publication and will be formally accepted for publication once it meets all outstanding technical requirements.

Kind regards,

Jisheng Liu, Ph.D.

Academic Editor

PLOS One

---

## [Editor Report · Acceptance letter]

PONE-D-25-38209R2

PLOS One

Dear Dr. Ren,

I'm pleased to inform you that your manuscript has been deemed suitable for publication in PLOS One. Congratulations! Your manuscript is now being handed over to our production team.

Kind regards,

on behalf of

Professor Jisheng Liu

Academic Editor

PLOS One